# Novel Disk Diffusion Assay on Magnesium Oxalate Agar To Evaluate the Susceptibility of *Yersinia pestis* to Type III Secretion System Inhibitors

Sukriti Prashar,[a] Miguel Portales Guemes,[a] Poorandai Shivbaran,[a] Eugenia Jimenez Alvarez,[a] Christopher Soha,[a] Samir Nacer,[a] Michael McDonough,[a] Gregory V. Plano,[b] Julie Torruellas Garcia[a]

[a]Department of Biological Sciences, Halmos College of Arts and Sciences, Nova Southeastern University, Fort Lauderdale, Florida, USA
[b]Department of Microbiology and Immunology, University of Miami, Miller School of Medicine, Miami, Florida, USA

**ABSTRACT** Current methods for screening small molecules that inhibit the plasmid pCD1-encoded *Yersinia pestis* type III secretion system (T3SS) include lengthy growth curves followed by multistep luminescence assays or Western blot assays to detect secretion, or lack thereof, of effector proteins. The goal of this research was to develop a novel disk diffusion assay on magnesium oxalate (MOX) agar as a simple way to evaluate the susceptibility of *Y. pestis* to type III secretion system inhibitors. MOX agar produces distinct *Y. pestis* growth characteristics based on the bacteria's ability or inability to secrete effector proteins; small, barely visible colonies are observed when secretion is activated versus larger, readily visible colonies when secretion is inhibited. Wild-type *Y. pestis* was diluted and spread onto a MOX agar plate. Disks containing 20 $\mu$l of various concentrations of imidocarb dipropionate, a known *Y. pestis* T3SS inhibitor, or distilled water (dH$_2$O) were placed on the plate. After incubation at 37°C for 48 h, visible colonies of *Y. pestis* were observed surrounding the disks with imidocarb dipropionate, suggesting that T3S was inhibited. The diameter of the growth of colonies surrounding the disks increased as the concentration of the T3SS inhibitor increased. Imidocarb dipropionate was also able to inhibit *Y. pestis* strains lacking effector Yops and Yop chaperones, suggesting that they are not necessary for T3S inhibition. This disk diffusion assay is a feasible and useful method for testing the susceptibility of *Y. pestis* to type III secretion system inhibitors and has the potential to be used in a clinical setting.

**IMPORTANCE** Disk diffusion assays have traditionally been used as a simple and effective way to screen compounds for antibacterial activity and to determine the susceptibility of pathogens to antibiotics; however, they are limited to detecting growth inhibition only. Consequently, antimicrobial agents that inhibit virulence factors, but not growth, would not be detected. Therefore, we developed a disk diffusion assay that could detect inhibition of bacterial virulence factors, specifically, type III secretion systems (T3SSs), needle-like structures used by several pathogenic bacteria to inject host cells with effector proteins and cause disease. We demonstrate that magnesium oxalate (MOX) agar can be used in a disk diffusion assay to detect inhibition of the T3SS of *Yersinia pestis*, the causative agent of bubonic plague, by small-molecule inhibitors. This assay may be useful for screening additional small molecules that target bacterial T3SSs or testing the susceptibility of patient-derived samples to drugs that target T3SSs.

**KEYWORDS** disk diffusion assay, *Yersinia pestis*, type III secretion systems, type III secretion system inhibitors, magnesium oxalate agar, imidocarb dipropionate, small molecule inhibitors

Address correspondence to Julie Torruellas Garcia, jg1511@nova.edu.

**Y**ersinia pestis, the causative agent of bubonic plague, is a Gram-negative bacterium that employs a type III secretion system (T3SS). The T3SS is a multiprotein complex organized in a needle-like structure specifically designed to export proteins from the

bacterial cytoplasm into a host cell (1). T3SSs are employed by many other bacterial pathogens, such as *Escherichia coli*, *Pseudomonas* spp., *Salmonella* spp., and *Chlamydia* spp. Though the activities and targets of the effector proteins that are injected into host cells differ between bacteria, the structure of the T3SS mechanism shows remarkable similarity between different species (2). Removal of the T3SS reduces the ability of a pathogen to evade host defenses and renders the bacterium avirulent (3). With the growing antibiotic resistance crisis, there is a need for discovery and development of new antibiotics and antivirulence agents. The similarity in structure and function of bacterial T3SSs and the reliance of many pathogens on the T3SS for virulence makes these secretion systems an attractive target for the development of new drugs.

Several methods have been utilized to screen small molecules for their ability to inhibit T3S in *Yersinia* spp. These include exposing *Y. pestis* to the small molecules and then conducting growth curves to determine whether growth was affected, Western blot assays to determine whether T3S effector proteins were secreted and to what extent, and translocation assays to determine whether the bacteria were able to use a T3SS to inject effector proteins directly into cultured cells (4).

In this study, we demonstrate the use of a novel disk diffusion assay to test the ability of the small molecules to inhibit the *Y. pestis* T3SS. The unique regulation of the T3SS in *Y. pestis* makes the bacterium useful for developing methods for screening small molecules that interfere with its regulation. The *Y. pestis* pCD1-encoded T3SS can be controlled *in vitro* using calcium. At 37°C, high levels of $Ca^{2+}$ block T3S while low levels induce secretion of *Yersinia* outer proteins (Yops) (5). Interestingly, *Y. pestis* undergoes growth restriction when the T3SS is induced, which our disk diffusion assay takes advantage of (4). Strains of *Y. pestis* with a T3SS that is regulated by $Ca^{2+}$ are referred to as calcium dependent. *Y. pestis* mutants that interfere with the calcium regulation of T3S exhibit one of two phenotypes. The first is the calcium-independent phenotype; mutants exhibiting this phenotype are not able to secrete Yops regardless of the calcium concentration and therefore lack growth restriction. The second is the calcium-blind phenotype; mutants with this phenotype secrete Yops regardless of the calcium concentration and undergo growth restriction (6).

The growth restriction that accompanies Yop secretion can be detected using magnesium oxalate (MOX) agar medium. Magnesium oxalate serves as a calcium chelator that sequesters any calcium in the medium, leading to the induction of *Yersinia* T3S at 37°C. Therefore, when grown on MOX agar and incubated at 37°C, *Yersinia* will form tiny, barely visible colonies if the T3SS is induced and larger, readily visible colonies if the T3SS is inhibited (6, 7). MOX agar has been used to screen *Yersinia* isolates from patients to determine if they are pathogenic or to screen for mutations that affect the T3SS (6, 8). Here, a novel use for MOX agar in a disk diffusion assay is described.

Disk diffusion assays are traditionally used to detect antibiotic activity and are evaluated by the appearance of a zone of inhibition of bacterial growth surrounding the disk containing the antibacterial compound; however, they are limited in their ability to detect other types of antimicrobial properties not related to growth inhibition, such as inhibition of virulence factors. In this study, a novel disk diffusion assay using MOX agar was developed to test the susceptibility of *Y. pestis* to T3SS inhibitors.

## RESULTS

***Y. pestis* growth on magnesium oxalate agar.** MOX agar contains sodium oxalate, a calcium chelator, which provides a low-calcium environment. *Y. pestis* undergoes growth restriction when grown in a low-calcium environment at 37°C (9). This growth restriction is correlated with active type III secretion. At room temperature, *Y. pestis* does not express pCD1-encoded T3SS genes and therefore does not undergo growth restriction regardless of whether it is grown in a low-calcium environment (10). Therefore, MOX agar is an ideal agar to use for differentiating between strains of *Y. pestis* with and without a functional T3SS. In this study, the recipe for MOX agar was modified for use with *Y. pestis*. In order to demonstrate that this modified MOX agar can be used to

RT    37°C

*wt*

*ΔyscF*

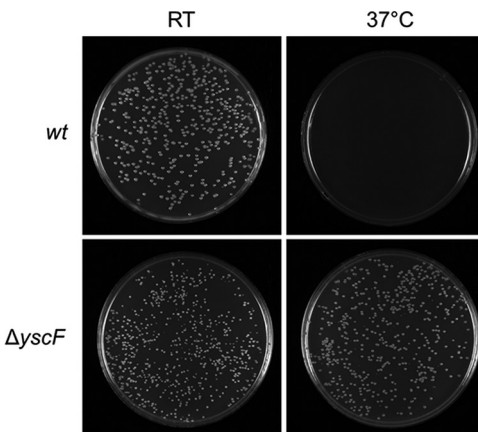

**FIG 1** Growth characteristics of *Yersinia pestis* wild-type (*wt*) and *ΔyscF* strains on MOX agar incubated at room temperature (RT) or 37°C for 48 h. Colonies of the wt are seen growing on MOX agar incubated at RT but not at 37°C, the temperature at which type III secretion is induced (top). Colonies of the *ΔyscF* strain, which does not have a functional type III secretion system, are seen growing on the MOX agar incubated at RT and at 37°C (bottom).

detect functional *Y. pestis* type III secretion based on differential growth characteristics, wild-type (wt) and *ΔyscF* strains were diluted and plated on MOX agar and incubated at room temperature (RT) or 37°C for 48 h. Colonies of the *Y. pestis* wt strain, which contain a functional T3SS, grew on MOX agar at RT; therefore, no growth restriction occurred (Fig. 1). No *Y. pestis* wt colonies could be seen on the MOX agar incubated at 37°C; therefore, growth restriction occurred, suggesting that a functional T3SS is present (Fig. 1). After further incubation at RT for 48 h, colonies could be seen, indicating that viable bacteria were present on the plate (data not shown). *Y. pestis ΔyscF* lacks the T3SS needle protein, resulting in a nonfunctional T3SS. Colonies of *Y. pestis ΔyscF* can be seen growing on MOX agar at both RT and 37°C, and no growth restriction occurred (Fig. 1). Overall, these results indicate that MOX agar is a differential agar that can be used to distinguish between *Y. pestis* strains that are able to secrete effector proteins through their T3SS from those that are not able to.

**A disk diffusion assay on MOX agar can be used to detect inhibition of *Y. pestis* type III secretion.** The differential growth morphologies displayed by various strains of *Y. pestis* on MOX agar provide a culture-based screening method to detect whether a functional T3SS is present. In order to determine if MOX agar could be used to screen small-molecule inhibitors of the *Y. pestis* T3SS, a disk diffusion assay was developed using imidocarb dipropionate, a known *Y. pestis* T3SS inhibitor. When wild-type *Y. pestis* was exposed to disks containing 20 $\mu$l of various concentrations of imidocarb dipropionate on MOX agar and incubated at 37°C, colonies were seen growing around the disk (Fig. 2A, right). This growth was dose dependent, as the diameter of the zone of T3S inhibition increased as the concentration of imidocarb dipropionate increased (Fig. 2B). No growth was seen around the control disk containing distilled water (dH$_2$O), suggesting that *Y. pestis* secreted effector proteins and as a result underwent growth restriction (Fig. 2A, left). Growth of *Y. pestis* was present on MOX agar plates incubated at RT but not on plates incubated at 37°C, indicating that the bacteria were viable and exhibiting the growth phenotype of *Y. pestis* with a functional T3SS (data not shown). These results suggest that a disk diffusion assay on MOX agar can be used to determine the susceptibility of *Y. pestis* to a known T3S inhibitor, imidocarb dipropionate.

**Inhibition of *Y. pestis* type III secretion by imidocarb dipropionate is not dependent on Yop effector proteins.** Although imidocarb dipropionate has been shown to inhibit *Y. pestis* T3S, its mode of inhibition is unknown. In order to begin to elucidate the mechanism of action of imidocarb dipropionate on T3S inhibition, our novel disk diffusion assay on MOX agar was performed with several *Y. pestis* mutant strains, including *Y. pestis* pCD1 Δ1234, lacking the effector proteins YopE, YopH, YopJ, YopM, and YopT and

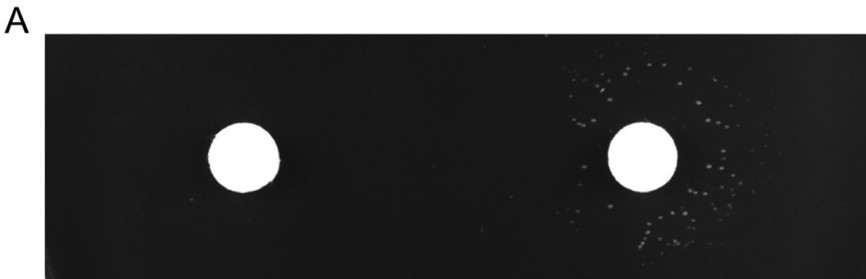

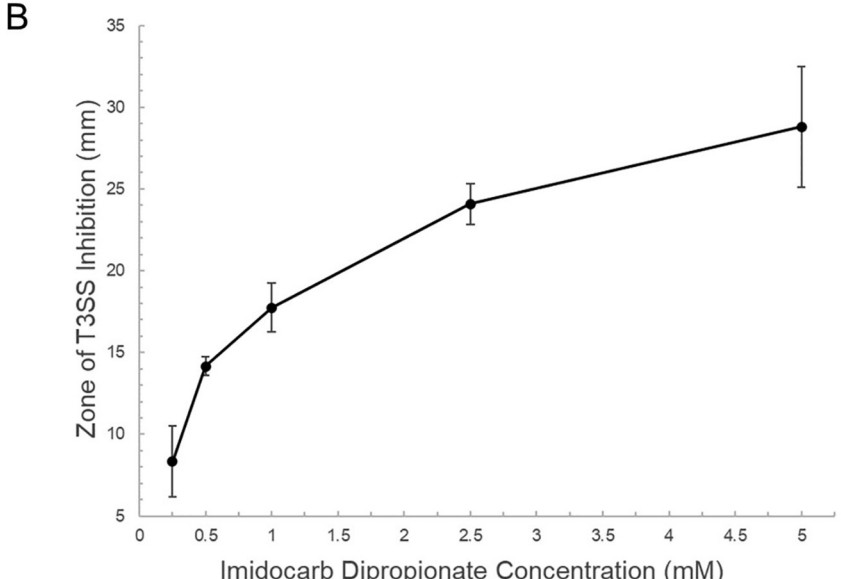

**FIG 2** Detection of *Y. pestis* type III secretion system inhibition by various concentrations of imidocarb dipropionate using a disk diffusion assay on MOX agar. (A) *Y. pestis* wt was plated onto MOX agar, and a disk with 20 μl of dH$_2$O (left) and one with 20 μl of 2.5 mM imidocarb dipropionate (right) were placed onto the plates. After incubation at 37°C for 48 h, colonies are seen growing around the disk containing imidocarb dipropionate but not around the control disk containing dH$_2$O, demonstrating inhibition of type III secretion by imidocarb dipropionate. (B) Dose-dependent effect of imidocarb dipropionate on inhibition of *Y. pestis* T3SS. The disk diffusion assay on MOX agar with *Y. pestis* wt was performed with 0.25 mM, 0.5 mM, 1 mM, 2.5 mM, 5 mM, and 10 mM imidocarb dipropionate. The zone of T3SS inhibition was determined by measuring the diameters of the growth of colonies surrounding the disks containing imidocarb dipropionate (in millimeters). Means and standard deviations from four replicates are shown.

the chaperones SycE, SycT, and YpkA and exhibiting a calcium-dependent phenotype (11); the Δ*yopN*x strain, lacking the T3SS regulatory protein YopN and Pla protease and exhibiting a calcium-blind phenotype (12); and YscF D46A, containing a point mutation in the YscF needle protein that leads to a calcium-blind phenotype (13). T3S in all three strains was inhibited by imidocarb dipropionate, as demonstrated by the growth seen on the MOX agar around the disk containing 2.5 mM imidocarb dipropionate but not around the disk containing dH$_2$O only (Fig. 3). Interestingly, imidocarb dipropionate was able to inhibit the calcium-blind Δ*yopN*x and YscF D46A strains, suggesting that the manner in which imidocarb dipropionate inhibits *Y. pestis* T3S differs from the manner in which high levels of calcium inhibit T3S (Fig. 3). Together, these results suggest that imidocarb dipropionate does not require the effector Yops, YopN, Yop chaperones or the Pla protease to inhibit T3S.

## DISCUSSION

The purpose of this study was to develop a novel disk diffusion assay to demonstrate that MOX agar can be used to evaluate the susceptibility of *Y. pestis* to T3SS inhibitors, such as imidocarb dipropionate. We took advantage of the growth restriction that accompanies *Y. pestis* T3S activity induced by low levels of calcium to devise

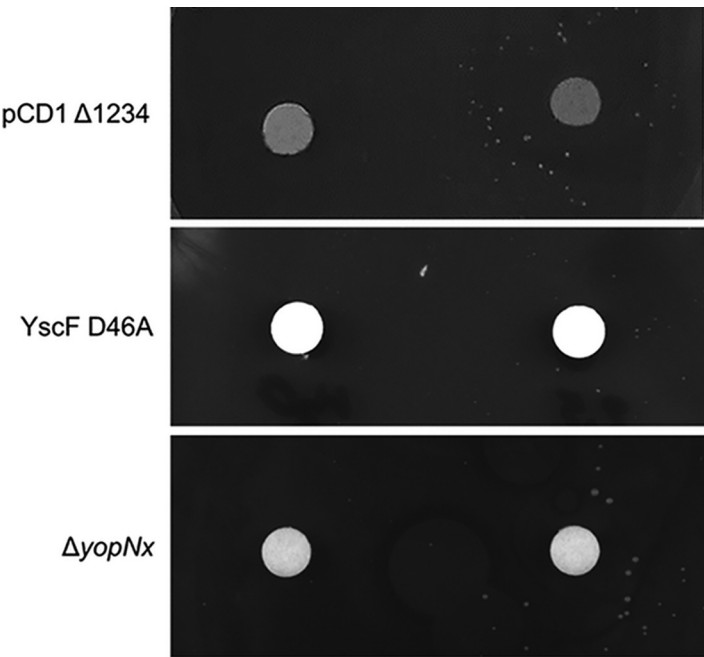

**FIG 3** Disk diffusion assay on MOX agar showing imidocarb dipropionate inhibits type III secretion of *Y. pestis* pCD1 Δ1234, YscF D46A, and Δ*yopN*x mutant strains. *Y. pestis* pCD1 Δ1234 lacking YopE, YopH, YopJ, YopM, YopT, SycE, SycT, and YpkA showed no colonies growing around the disk containing 20 μl of dH₂O (left); however, colonies are seen growing around the disk containing 2.5 mM imidocarb dipropionate (right), suggesting that type III secretion was inhibited (top). Similar results are seen for the YscF D46A strain, which contains a mutation in the T3SS needle protein resulting in constitutive type III secretion (middle), and the Δ*yopN*x strain, which lacks YopN, a protein involved in the regulation of type III secretion whose deletion leads to constitutive secretion, and also lacks the *pla* protease (bottom). Therefore, YopE, YopH, YopJ, YopM, YopT, SycE, SycT, YpkA, YopN, *pla* protease, and YscF amino acid residue 46 are not necessary for imidocarb dipropionate to inhibit *Y. pestis* type III secretion.

an alternative screening technique. Since MOX agar contains sodium oxalate, a calcium chelator, we were able to use it to create a low-calcium environment that resulted in differential growth of *Y. pestis* based on its ability to secrete and to restrict growth (Fig. 1). This study is the first to demonstrate the novel use of MOX agar in a disk diffusion assay to screen for *Y. pestis* T3SS inhibitors. We initially used one known *Y. pestis* T3SS inhibitor, imidocarb dipropionate, that was commercially available to show proof of concept for our novel disk diffusion assay (Fig. 2). Future studies will include the use of additional T3SS inhibitors.

Unlike traditional disk diffusion assays that begin with spreading the test strain on the plate to create a lawn of growth, the disk diffusion assay on MOX agar requires that the bacteria form single colonies that are uniformly dispersed and sufficiently distant from one another. Too few colonies plated could lead to false-negative results, since there may not be colonies present around the disk containing the inhibitor. Therefore, it is suggested that when a negative result is seen, the plates should then be further incubated at RT to ensure that the lack of colonies around the disk was not due to the lack of bacteria around the disk. Conversely, starting with too many bacteria on the plate could lead to a potential false positive, as the bacteria will grow normally at 37°C even under secretion-inducing conditions when they are too crowded. To circumvent this issue, positive and negative controls should be used.

There are some limitations of this assay. The main limitation of this assay is that it will not detect T3S inhibitors that do not reverse growth restriction associated with *in vitro* activation of secretion. It may also detect false positives in the case of compounds that reverse growth restriction but do not actually inhibit secretion. Therefore, results should be confirmed by Western blotting to ensure that Yop secretion was, in fact, inhibited. A benefit

**TABLE 1** Construction of *Yersinia pestis* strains used in this study

| *Y. pestis* strain | Construction and properties | Reference |
| --- | --- | --- |
| KIM5-3001 (wt) | Sm$^r$ pCD1 pPCP1 pMT1 | 22 |
| KIM8-3002.7 (Δ*yopN*x) | Sm$^r$ pCD1 Δ*yopN* pPCP1$^-$ (Pla$^-$) pMT1 | 23 |
| KIM8 Δ1234 (pCD1 Δ1234) | Sm$^r$ Km$^r$ pCD1 Δoperon 1, 2, 3, 4, pPCP1$^-$ (Pla$^-$) pMT1 | 13 |
| KIM5-3001.P61(Δ*yscF*) | Sm$^r$ pCD1 (Δ*yscF*) pPCP1 pMT1 | 24 |
| KIM5-3001.P39-F4 (YscF D46A) | Sm$^r$ pCD1 (Δ*sycE-yopE*::Km YscF D46A) pPCP1 pMT1 | 24 |

to using this method is that since the small-molecule inhibitors diffuse through the agar, a concentration gradient is achieved without having to make and test multiple concentrations of the compound individually. However, this assay will not be appropriate for all types of compounds, since some may not be able to sufficiently diffuse through the agar medium, such as water-insoluble compounds (14).

Although several *Y. pestis* T3SS inhibitors have been identified, only a few protein targets of these inhibitors have been determined (15). Some small molecules have been discovered to inhibit the *Y. pestis* T3SS by altering gene expression (16) or by targeting a specific component of the T3S apparatus, such as the translocon (17), Yop effector proteins (10, 18), or injectisome assembly (19). Analyzing the various effects of different inhibitors will expand our understanding of the regulation of T3S and the diversity of T3SS inhibitors. As demonstrated in Fig. 3, our novel disk diffusion assay on MOX agar may be used with various mutant strains to narrow down and eventually pinpoint the molecular targets of T3SS inhibitors. Pan et al. observed that differential inhibition of Yops was exhibited by imidocarb dipropionate and that it inhibits the translocation of YopE into HeLa cells (10). However, it is unclear whether the reduction in Yop secretion and translocation was due to decreased gene expression. We showed that the Yops were not required for imidocarb dipropionate to inhibit T3S (Fig. 3). Interestingly, imidocarb dipropionate was able to inhibit *Y. pestis* strains that exhibit a calcium-blind phenotype. Although the exact mechanism by which calcium blocks T3S is unknown, these results suggest that imidocarb dipropionate uses a mechanism other than calcium to inhibit secretion. It is possible that imidocarb dipropionate affects gene expression or that it targets the T3SS apparatus to interfere with secretion.

Antibiotic-resistant bacteria are rapidly emerging worldwide, threatening the efficacy of current antibiotics; therefore, there is a need for new antibiotics and antivirulence agents. T3SSs are an attractive target due to their essential role in causing disease. Our novel disk diffusion assay could potentially aid in the discovery of new drugs that target T3S. MOX agar has never been used to screen compounds that inhibit T3SSs; however, we have shown that it can be used for this purpose (Fig. 2). Ideally, the inhibitors discovered using this assay could be useful in the treatment of other bacterial pathogens that utilize a T3SS. Pan et al. (10) demonstrated that imidocarb dipropionate inhibited T3S in both *Y. pestis* and enteropathogenic *E. coli*; therefore, it is reasonable to predict that T3SS inhibitors discovered using our novel disk diffusion assay could potentially also target other pathogens that utilize a T3SS (8). Since imidocarb dipropionate (brand name, Imizol) is FDA approved for use to treat a protozoal infection in dogs, known as babesiosis, caused by transfer of the *Babesia* pathogen to pets through tick bites, it would be interesting to investigate its potential off-label use as a treatment for *Y. pestis* infections in an animal model (20). As drugs that target T3S become available for medical use, we envision that this assay could be useful as a simple and inexpensive way to evaluate patient-derived samples to determine their level of susceptibility to these drugs, analogous to how the Kirby-Bauer test is used to evaluate susceptibility to traditional antibiotics (21).

## MATERIALS AND METHODS

**Bacterial strains and growth conditions.** Bacterial strains used in this study are listed in Table 1. *Y. pestis* strains were routinely grown on tryptic soy agar (TSA) plates (Difco) at room temperature and stored at 4°C. Antibiotics were routinely added to the TSA plates to ensure proper growth at concentrations of 50 $\mu$g/ml (streptomycin and/or kanamycin).

**Preparation of magnesium oxalate agar.** To make 1 liter of MOX agar, 34 g of tryptic soy agar (Difco) was added to 840 ml of dH$_2$O and autoclaved, with a magnetic stir bar, at 121°C for 20 min. Next, 100 ml of 0.25 M MgCl$_2$ and 100 ml of 0.25 M Na$_2$C$_2$O$_4$ were prepared and autoclaved at 121°C for 20 min. Then, 80 ml 0.25 M MgCl$_2$ and 80 ml 0.25 M Na$_2$C$_2$O$_4$ were added to the TSA and mixed by gently stirring on a stir plate. A 25-ml portion of this mixture was added to each petri dish. Plates were left at room temperature for 24 h to dry and then stored in the refrigerator until use.

**Growth and disk diffusion assay on MOX agar.** Sterile dH$_2$O was inoculated with *Y. pestis* KIM5-3001 wild type (wt) or KIM5-3001.P61 (ΔyscF), and the density of the suspension was adjusted to a 0.5 McFarland turbidity standard (∼$1.5 \times 10^8$ CFU/ml). The suspension was then serially diluted in 10-fold increments until a $10^{-4}$ dilution was reached. One hundred microliters of the $10^{-4}$ dilution was plated and spread onto MOX agar plates supplemented with 50 μg/ml streptomycin to achieve a final concentration of 300 to 500 colonies. Plates were incubated at RT for 48 h as a control for viability or at 37°C for 48 h to ensure that growth restriction was detected. For the disk diffusion assay, 20 μl of 0.25 mM, 0.5 mM, 1 mM, 2.5 mM, 5 mM, or 10 mM imidocarb dipropionate (Sigma) was inoculated onto a 6-mm disk and placed onto the MOX agar plate preinoculated with wt *Y. pestis*. A 6-mm disk inoculated with 20 μl of sterile dH$_2$O was used as a negative control. These plates were incubated at 37°C for 48 h. The zone of T3SS inhibition was determined by measuring the diameters of the growth of colonies surrounding the disks, in millimeters. Means and standard deviations from four replicates were calculated.

**Determining the molecular target of imidocarb dipropionate.** The method for the disk diffusion assay on MOX agar was used with a few modifications. Sterile phosphate-buffered saline (PBS) was used in place of dH$_2$O for *Y. pestis* mutant strain inoculations. Sterile PBS was inoculated with *Y. pestis* KIM8 (pCD1 Δ1234), KIM8-3002.7 (ΔyopNx), or KIM5-3001.P39-F4 (YscF D46A) to reach a 0.5 McFarland turbidity standard (∼$1.5 \times 10^8$ CFU/ml). The optical density at 600 nm (OD$_{600}$) was measured to ensure consistent turbidity across strains. The suspension was then serially diluted in PBS in 10-fold increments until a $10^{-3}$ dilution was reached. One hundred microliters of the $10^{-3}$ dilution was plated and spread onto three MOX agar plates supplemented with 50 μg/ml streptomycin or 50 μg/ml of kanamycin. One plate was incubated at room temperature for 48 h as a viability control, and one plate was incubated at 37°C for 48 h to ensure that growth restriction was detected. On the third plate, 20 μl of 1 mM imidocarb dipropionate (Sigma) was inoculated onto a 6-mm disk and placed on the MOX agar plate preinoculated with a *Y. pestis* mutant strain. Twenty microliters of sterile PBS was used as a negative control. The plate was incubated at 37°C for 48 h, after which the plates were observed for growth around the disks.

## ACKNOWLEDGMENT

This project was funded by a Nova Southeastern University President's Faculty Research and Development Grant (PFRDG), index 335407. The funders had no role in study design, data collection and interpretation, or the decision to submit the work for publication.

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
