## [Reviewer comments · Microbiology Spectrum]

**Microbiology
Spectrum**

Novel Disk Diffusion Assay on Magnesium Oxalate Agar to Evaluate the Susceptibility of *Yersinia pestis* to Type III Secretion System Inhibitors

Sukriti Prashar, Miguel Portales Guemes, Poorandai Shivbaran, Eugenia Jimenez Alvarez, Christopher Soha, Samir Nacer, Michael McDonough, Gregory Plano, and Julie Torruellas Garcia

Corresponding Author(s): Julie Torruellas Garcia, Nova Southeastern University

Review Timeline:

Submission Date:

April 19, 2021

Accepted:

April 20, 2021

Editor: Christina Cuomo

Reviewer(s): The reviewers have opted to remain anonymous.

Transaction Report:

DOI: <https://doi.org/10.1128/Spectrum.00005-21>

April 20, 2021

Dr. Julie Torruellas Garcia
Nova Southeastern University
Biological Sciences
3301 College Ave.
Parker 388
Fort Lauderdale, FL 33314

Re: Spectrum00005-21 (Novel Disk Diffusion Assay on Magnesium Oxalate Agar to Evaluate the Susceptibility of *Yersinia pestis* to Type III Secretion System Inhibitors)

Dear Dr. Julie Torruellas Garcia:

Thank you for submitting your manuscript to Microbiology Spectrum. Based on editorial evaluation, I have accepted this paper, and recommend that it should be showcased as a "Methods and Protocols" paper. Please confirm that this acceptable when approving final files.

Your manuscript has been accepted, and I am forwarding it to the ASM Journals Department for publication. You will be notified when your proofs are ready to be viewed.

Sincerely,

Christina Cuomo
Editor, Microbiology Spectrum
